# Strength Characteristics of Sand–Silt Mixtures Subjected to Cyclic Freezing-Thawing-Repetitive Loading

**DOI:** 10.3390/s20185381

**Published:** 2020-09-20

**Authors:** Jong-Sub Lee, Jung-Doung Yu, Kyungsoo Han, Sang Yeob Kim

**Affiliations:** 1School of Civil, Environmental and Architectural Engineering, Korea University, 145, Anam-ro, Seongbuk-gu, Seoul 02841, Korea; jongsub@korea.ac.kr (J.-S.L.); noorung2@korea.ac.kr (J.-D.Y.); 2Lyles School of Civil Engineering, Purdue University, 550, Stadium Mall Drive, West Lafayette, IN 47907, USA; han599@purdue.edu

**Keywords:** freezing-thawing-repetitive loading, sand–silt mixtures, silt fraction, unconfined compressive strength, volumetric unfrozen water content

## Abstract

Daily freezing-thawing-repetitive loading is a critical factor affecting soil stability. This study assesses the strength of sand–silt mixtures with various silt fractions (SFs) subjected to cyclic freezing-thawing-repetitive loading. Specimens with SF of 0–100% were prepared with a fixed relative density of 60%. The number of repetitive loadings (N) was 1, 100, and 1000 for each specimen with different SFs. After three cycles of freezing-thawing-repetitive loading, the specimens were frozen at −5 °C for the uniaxial compression test. Test results show that the change in relative density (∆D_r_) increases with the increase in SF up to 30% and decreases as SF increases beyond 30% owing to the change in the void ratio. The volumetric unfrozen water content (θ_u_) increases with the increase in both SF and N owing to the effect of the physicochemical characteristics of soils on small voids. Unconfined compressive strength of sand-dominant mixtures (SF ≤ 30%) is reinforced by ∆D_r_. By contrast, for silt-dominant mixtures (SF > 30%), the unconfined compressive strength decreases with the increase in θ_u_ and N due to lubricant role and sands dispersion. Thus, the effects of SF and N should be considered for sand–silt mixtures that have a probability to undergo cyclic freezing-thawing-repetitive loading.

## 1. Introduction

In numerous cold regions, the ground may undergo daily as well as seasonal freeze-thaw cycles in the active layer [1,2]. For daily frozen-thawed ground, the soils are subjected to several freeze-thaw cycles, as they can freeze at night and thaw in the daytime. In the case of the daily frozen-thawed ground underneath a road or railway, the soils that thaw during the day are exposed to repetitive loading from dynamic vehicle loads. As cyclic freezing-thawing-repetitive loading may change the soil structure, the strength of soils should be investigated to maintain soil stability [3].

Freeze-thaw cycles generally induce particle rearrangement through volume expansion during freezing and contraction during thawing, which changes the void ratio and permeability [4,5,6]. The alteration of soil indices and properties may reduce the strength of soils [7]. Likewise, repetitive loading may cause both volume contraction and expansion with respect to the initial packing density and fine fraction [8,9]. In addition, repetitive mechanical loading changes the engineering parameters, such as reduced strength due to excess pore water pressure [10]. Thus, the effect of freezing-thawing-repetitive loading on the daily frozen-thawed ground underneath a road or railway should be considered in geotechnical design.

Previous studies have explored the change in void ratio and strength parameters after freeze-thaw cycles for sand, silt, or clay [11,12]. The effects of repetitive loading on the void ratio evolution and engineering parameters of pure soils have also been widely discussed [13,14]. Several studies have revealed the effects of cyclic freezing-thawing-repetitive loading on soil structure alteration [3]. In particular, the strength parameter can significantly change according to the fine fraction for coarse-fine mixtures [15]. However, previous studies on the strength of soils subjected to cyclic freezing-thawing-repetitive loading have dealt with each type of soils. Thus, the strength of sand–silt mixtures, which represent coarse-fine mixtures, subjected to cyclic freezing-thawing-repetitive loading was characterized in this study. The analyses of the test results cover the effect of silt fraction (SF) on the strength of sand–silt mixtures as well as the influence of cyclic freezing-thawing-repetitive loading.

The objective of this study is to explore the effect of cyclic freezing-thawing-repetitive loading on the strength of sand–silt mixtures with various SFs. First, the preparation of sand–silt mixtures with SF from 0% to 100% is described, and a repetitive loading system containing the time-domain reflectometry (TDR) mold is introduced. Next, the volume change and volumetric water content during three cyclic freezing-thawing-repetitive loadings are measured, and the unconfined compressive strength is estimated after the last freezing. Subsequently, the relationship between the change in relative density (∆D_r_) and volumetric unfrozen water content (θ_u_) was analyzed. Finally, the relationships between ∆D_r_ and θ_u_ and the unconfined compressive strength of sand-dominant (SF ≤ 30%) and silt-dominant (SF > 30%) mixtures are discussed to estimate and compare the influence of ∆D_r_ and θ_u_.

## 2. Materials and Methods

### 2.1. Sand–Silt Mixtures

Sand particles with sizes ranging from 0.25 to 0.60 (passed through sieve #30 and retained on sieve #60) and silt with particle size less than 0.075 mm (passed through sieve #200) were prepared. The specific gravities of the prepared sand and silt were 2.62 and 2.71, respectively. Silt was mixed into sand with SF = 0, 10, 30, 50, 70, and 100% by weight (W_silt_/W_sand_ × 100%). The maximum and minimum void ratios of each sand–silt mixture with different SF were determined based on the American Society for Testing and Materials (ASTM) D4253 and D4254 standards [16,17], and they varied with the SF owing to the role of fine particles [18] as shown in Figure 1. Note that the maximum void ratio presents the loosest packing condition (i.e., minimum density state), and the minimum void ratio displays the densest packing condition (i.e., maximum density state). Thus, the maximum void ratio can be obtained by pouring the specimen from the spout as loosely as possible, while the minimum void ratio can be attained by compacting and vibrating the specimen in the mold. In addition, the silts, which acted as fine particles in this study, filled up the voids to a threshold SF of 30% and dispersed the sands, which are the coarse particles, beyond the SF of 30%. As the relative density (D_r_) of sand–silt mixtures was fixed at 60%, the initial void ratio varied with respect to the SF. Note that the D_r_ is a ratio of the difference between maximum and any given void ratios to that between maximum and minimum void ratios [17]. The degree of saturation (S) was fixed at 15% for all mixtures. The prepared sand–silt mixtures were set into the floating oedometer cell by tamping in five layers with an equivalent compaction number and energy. After the preparation of sand–silt mixtures, the floating oedometer cell was placed in the repetitive loading system, which was then placed in a freezing chamber to apply cyclic freezing-thawing-repetitive loading. Note that the freezing temperature is fixed at −5 °C to minimize the temperature effect and differentiate unfrozen water content affected by silt fraction.

### 2.2. Repetitive Loading System

The stress-controlled loading system was designed to subject the sand–silt mixtures to repetitive loading. The repetitive loading system consisted of a pneumatic cylinder, loading frame, floating oedometer cell with a TDR probe and a linear variable displacement transducer (LVDT), as shown in Figure 2. The pneumatic cylinder was operated by a pneumatic valve and controller, which activated the sinusoidal stress amplitude of 50 kPa monitored by a pressure transducer for repetitive loading. The loading frame was made of stainless steel and assembled by double-bolting to prevent other deformation factors during repetitive loading. The floating oedometer cell was made of mono-cast nylon to minimize the electrical interference of the TDR probe, which was incorporated into the cell. The diameter, height, and thickness of the floating oedometer cell was 50, 100, and 10 mm, respectively. The LVDT was installed at the top cap and measured the change in specimen height during cyclic freezing-thawing-repetitive loading before the uniaxial compression test. The measured specimen height was automatically saved in a computer via the data logger.

The test procedure mainly consists of specimen preparation, cyclic freezing-thawing-repetitive loading, and uniaxial compression testing as listed in Table 1. All sand–silt mixtures were prepared with a D_r_ of 60% as the initial condition and equivalently underwent freezing, thawing, and repetitive loading. Three freezing-thawing-repetitive loading cycles were applied in sequence. The repetitive loadings, with numbers of 1, 100, and 1000, were subjected to sand–silt mixtures after each thawing. Note that the repetitive loading period was set as 12 s (frequency = 0.083 Hz) to minimize the dynamic effect on strain accumulation [19,20,21]. After the cyclic freezing-thawing-repetitive loading, the sand–silt mixture was frozen to conduct the uniaxial compression test.

### 2.3. Time-Domain Reflectometry Probe

The electromagnetic signals of the TDR probe have been widely used to estimate the volumetric water content of soils because the velocity of the electromagnetic wave changes with the relative permittivity of the surrounding material. The velocity of the electromagnetic waves was calculated from the travel time, which is expressed as follows:(1)$$\varepsilon_{r}={\left(\frac{c}{v}\right)}^{2}={\left(\frac{c\times \Delta t}{2L}\right)}^{2}$$
where *ε_r_* is the relative permittivity; *c* and *v* are the electromagnetic wave velocity in vacuum and the soil; and Δ*t* and *L* are the travel time of the electromagnetic wave in soil and the TDR probe length, respectively. The TDR probe manufactured in this study consisted of one central electrode that propagates and reflects electromagnetic waves; and two outer electrodes, which determine the electromagnetic field [22]. The length, width, and thickness of all electrodes were 80, 2, and 2 mm, respectively. The center-to-center distance between each electrode was 10 mm, as shown in Figure 2. The center electrode was soldered to the inner coaxial cable, and the outer electrodes were soldered to the outer coaxial cable to connect the time-domain reflectometer.

Previous studies have revealed that the relationship between the volumetric water content and relative permittivity can be expressed by a cubic polynomial equation [15,23,24]. In addition, as the relative permittivities of dry soils and ice are similar, this relationship has been used to estimate the unfrozen water content of frozen soils [25]. Typical TDR signals of soils before and after freezing are presented in Figure 3. The travel time (∆*t*) was determined by the difference between the first reflection time (*t*_0_) and second reflection time, which can be determined by adopting the point of intersection between two tangent lines, as shown in Figure 3. ∆*t*_1_ becomes ∆*t*_2_, as the partially saturated soil freezes because most of the water changes phase into ice, which has low relative permittivity. From the estimation of the relative permittivity (*ε_r_*) by substituting ∆*t*_2_ into Equation (1), volumetric unfrozen water content (*θ_u_*) has been commonly estimated using a cubic polynomial relationship as follows:(2)$$\theta_{u}=a\cdot \varepsilon_{r}{}^{3}+b\cdot \varepsilon_{r}{}^{2}+c\cdot \varepsilon_{r}+d$$
where *a*, *b*, *c*, and *d* have been experimentally obtained as shown in Table 2. The coefficients are determined as *a* = 0.95, *b* = −12.99, *c* = 64.48, and *d* = −95.04 through the calibration test.

## 3. Results

### 3.1. Volumetric Response

To estimate the volume change of sand–silt mixtures during cyclic freezing-thawing-repetitive loading, variations in specimen height were continuously measured. Typical measured heights of sand–silt mixtures are plotted in Figure 4. Figure 4a presents the variations in height of the sand–silt mixture with silt fraction (SF) of 70% subjected to repetitive loading (N) of 1, 100, and 1000. All the heights of sand–silt mixtures rapidly increase (i.e., volume expansion) during freezing and gradually decrease (i.e., volume contraction) during thawing. The soil particles moved apart owing to the volumetric expansion of ice in the voids during the phase change from water to ice and were progressively compacted by the particle rearrangement during the phase change from ice to water [28]. The increased N after thawing induced further contraction of sand–silt mixtures [29]. To clarify the volume contraction of sand–silt mixtures during repetitive loading, the specimen height variation under N = 100 loading in the first cycle is typically replotted with an enlarged scale of the x- and *y*-axis, as shown in Figure 4b. Repetitive loading causes volume contraction and tends to converge at N = 100. Note that most strain accumulation occurs within N = 100, and convergence begins beyond N = 100 [30].

To estimate the effect of SF on the volume change, the changes in height and relative density (D_r_) of sand–silt mixtures with respect to the SF are plotted in Figure 5. Figure 5a shows that for all repetitive loadings (N = 1, 100, and 1000), the change in the height of the sand–silt mixture decreases, as the SF increases to 30%, and increases beyond an SF of 30%. As the initial void ratios for all specimens were fixed at a D_r_ of 60%, the change in height of sand–silt mixtures varied according to the SF. The change in specimen height according to the initial void ratio is plotted in Figure 5b. Figure 5b shows that the variation in specimen height increases with the increase in the initial void ratio. Note that the change in relative density (∆D_r_) should be considered as well as the specimen height because the mechanical characteristics of sand–silt mixtures, such as strength, are significantly affected by the D_r_ rather than the void ratio [31]. Figure 5c shows that ∆D_r_ increases according to SF up to 30% and decreases when SF exceeds 30%. The available range of void ratios from the minimum to the maximum, which is the denominator for the calculation of D_r_, narrows up to an SF of 30% and widens beyond an SF of 30% (see Figure 1). Thus, the increasing and decreasing trend of ∆D_r_ is the opposite to that of the sand–silt mixture height in Figure 5a.

### 3.2. Volumetric Water Content

To estimate the variation in the volumetric water content of sand–silt mixtures during cyclic freezing-thawing-repetitive loading, the TDR signals after each freezing, thawing, and repetitive loading event were collected. Figure 6 shows the typical TDR signals of each phase after several freezing-thawing-repetitive loading cycles (Fn, Tn, and Rn) for the sand–silt mixture with SF of 70% subjected to N = 1000. The TDR signals move toward the left after freezing, which implies a decrease in relative permittivity. Note that the relative permittivity of water and ice are generally 80 and 5, respectively [32]. Subsequently, the TDR signals shift toward the right after thawing, compared with the signal before freezing. As the freeze–thaw cycle causes volume contraction (see Figure 4), at constant gravimetric water content, the estimated volumetric water content slightly increases [33]. The TDR signal moves toward the right after repetitive loading owing to the volume contraction. Consequently, the TDR signal generally moves toward the right as the sand–silt mixture undergoes increasing numbers of cyclic freezing-thawing-repetitive loading, which increases the measured volumetric water content.

To evaluate the effect of N on the volumetric water content of sand–silt mixtures with different SFs, the TDR signals of sand–silt mixtures with SF of 0% and 100% are plotted in Figure 7. Note that Nn denotes the number of repetitive loading cycles at each cycle, whereas Rn denotes the cycle number of repetitive loading. Figure 7a,b show that the TDR signals of the soils subjected to larger N move toward the right owing to compaction, which is also observed in Figure 6. However, the difference between the TDR signals of N = 1, 100, and 1000 for the sand–silt mixture with SF of 100% plotted in Figure 7b is greater than that for the clean sand in Figure 7a. The strain accumulation (i.e., volume contraction) of the sand–silt mixture subjected to repetitive loading increases as the SF increases [29]. The larger N, which induces a greater ∆D_r_ and increases the chances of pore water being affected by the surface of physicochemical soil particles, leads to a greater θ_u_ [34,35]. A number of previous studies have revealed that small voids induced more unfrozen water owing to the physicochemical characteristics of soils [15,26]. Free water in large voids is less likely to be affected by the surface of soil particles, which reveals the physicochemical characteristics. Thus, the difference between the TDR signals in Figure 7b is clearer than that in Figure 7a, even though the Nn was subjected to the same repetitive loading.

The θ_u_, which was measured at the fourth freezing phase after three cyclic freezing-thawing-repetitive loadings, was estimated from the TDR signals. The θ_u_ of sand–silt mixtures subjected to N = 1, 100, and 1000 with respect to the SF are plotted in Figure 8. Figure 8 shows that θ_u_ increases as N increases owing to the effects of compaction and physicochemical soil particles. In particular, the θ_u_ of sand–silt mixtures subjected to N = 1000 is slightly greater than that of N = 100, even though the sand–silt mixture was subjected to additional loading of N = 900. Most volume contractions due to repetitive loading generally occur within N = 100 (see Figure 4b). Thus, the difference between the estimated θ_u_ of sand–silt mixtures subjected to N = 1 and 100 is much greater than those subjected to N = 100 and 1000. Meanwhile, θ_u_ increases with an increase in SF. As increased SF decreases the voids in the sand–silt mixtures, more unfrozen water remained near the surface of soil particles, which revealed their physicochemical characteristics.

### 3.3. Unconfined Compressive Strength

The unconfined compressive strength was determined at the peak value of the unconfined compressive stress. For the comparison of the unconfined compressive strength of sand–silt mixtures subjected to different N, the unconfined compressive stress–strain curves of sand–silt mixtures with SFs of 30% and 70% and subjected to N = 1, 100, and 1000 are plotted in Figure 9. Figure 9a shows that the unconfined compressive strength increases with the increase in N for the sand–silt mixture with an SF of 30%. By contrast, the unconfined compressive strength of the sand–silt mixture with an SF of 70% decreases with the increase in N, as shown in Figure 9b. Fine particles in coarse-dominant mixtures generated active particle contacts in the dense skeleton of coarse particles, which induced an increase in strength; those in fine-dominant mixtures dispersed the coarse particles, which may reduce the strength [9,36]. Note that silt acts as fine particles while sand plays the role of coarse particles in this study. Thus, in the sand-dominant mixture (SF = 30%), the unconfined compressive strength increases with the increase in N, which induced a denser sand skeleton. In the silt-dominant mixture (SF = 70%), the unconfined compressive strength decreases with the increase in N, which leads to the dispersion of sand particles.

To clarify the effects of silt particles on sand-dominant and silt-dominant mixtures, the unconfined compressive stress–strain curves for sand–silt mixtures with SF from 0% to 100% subjected to N = 100 are plotted in Figure 10. Figure 10 shows that the unconfined compressive strength increases with the increase in SF up to 30% and decreases as the SF increased beyond 30%. The greater SF generates more active particle contacts for sand-dominant mixtures (SF ≤ 30%); and may disperse the sand particles and reduce the unconfined compressive strength for silt-dominant mixtures (SF > 30%), as shown in Figure 9. The unconfined compressive stress of sand-dominant mixtures rapidly decreases after the peak stress, while that of silt-dominant mixtures gradually decreases according to the axial strain. In other words, the sand-dominant mixtures exhibited strain-softening, which can occur in brittle materials. The behavior of silt-dominant mixtures became more strain-hardening, which is developed in ductile materials. Thus, the addition of silt particles turns brittle materials into ductile materials for frozen sand–silt mixtures.

To determine the effects of both SF and N on the mechanical characteristics of sand–silt mixtures, the unconfined compressive strength of sand–silt mixtures with SF from 0% to 100% subjected to N = 1, 100, and 1000 are plotted in Figure 11. The unconfined compressive strength increases as the SF increases up to 30% and decreases beyond SF of 30%. Previous studies have reported that the strength of sand–silt mixtures at a fixed relative density decreased up to an SF of 30% and increased beyond SF of 30% [15,26,37], which is the opposite of the trend in Figure 11. In this study, however, the additional cyclic freezing-thawing-repetitive loading induced a different ∆D_r_ according to the SF (see Figure 5c), which resulted in variations in unconfined compressive strength. The unconfined compressive strength increases with the increase in N at low SF but decreases as N increases at high SF. The greater N generated the active particle contact of fine particles at low SF (see Figure 9a). However, the greater N also dispersed the sand particles (see Figure 9b) and increased the unfrozen water content, which resulted in ductile failure behavior (see Figure 8) at a high SF. Thus, the repetitive loading may reinforce or weaken sand–silt mixtures depending on whether these are sand-dominant or silt-dominant, which correspond to coarse-dominant or fine-dominant for general coarse-fine mixtures.

## 4. Discussion

### 4.1. Volumetric Unfrozen Water Content vs. Change in Relative Density

To evaluate the effects of cyclic freezing-thawing-repetitive loading on the θ_u_ and ∆D_r_ of frozen sand–silt mixtures, the relationship between the θ_u_ and ∆D_r_ is plotted in Figure 12. The θ_u_ increases with the increase in SF (see Figure 8) because of the effects of the physicochemical characteristics of the particles. The ∆D_r_ increases as the SF increases up to 30% (see Figure 5c) owing to the narrower range of maximum and minimum void ratios (see Figure 1). The θ_u_ and ∆D_r_ increase with the increase in the number of repetitive loadings (see Figure 5c and Figure 8) because the unfrozen water that remains in the small voids is easily affected by the surface of physicochemical soil particles. Thus, the θ_u_ is linearly proportional to the ∆D_r_ of sand–silt mixtures with SF less than 30%.

The θ_u_ of sand–silt mixtures with SF greater than 30% tends to increase as the ∆D_r_ decreases or is fixed. The θ_u_ increases as SF increases owing to the effect of silt particles (see Figure 8), while ∆D_r_ decreases or is maintained owing to the wider range of maximum and minimum void ratios (see Figure 1). The greater number of repetitive loadings after thawing resulted in smaller voids before freezing, which caused more volume expansion during freezing (see Figure 4a) and decreased the ∆D_r_. Thus, the θ_u_ of sand–silt mixtures with SF beyond 30% subjected to repetitive loading increases as the ∆D_r_ decreases, while that of sand–silt mixtures that undergoes a single freeze-thaw cycle, i.e., N = 1, increases as the ∆D_r_ is maintained.

### 4.2. Unconfined Compressive Strength vs. Change in Relative Density

To determine the influence of the ∆D_r_ on the strength of sand–silt mixtures subjected to cyclic freezing-thawing-repetitive loading, the unconfined compressive strength versus the ∆D_r_ is plotted in Figure 13. Figure 13 shows that the unconfined compressive strength of sand–silt mixtures increases with the increase in N, as the SF increases up to 30% and tends to decrease as the SF increases beyond 30%, which represents the transition from sand-dominant to silt-dominant behavior [15]. As mentioned above, silt acts as fine particles while sand plays the role of coarse particles in this study. For the sand-dominant mixtures (SF ≤ 30%), the unconfined compressive strength increases with the increase in SF because the fine particles in coarse-dominant mixtures generate active particle contacts [9]. Furthermore, the unconfined compressive strength increases with the increase in N because of the increased active particle contacts in denser sand-dominant mixtures [38]. Thus, the unconfined compressive strength increases with the increase in ∆D_r_, which results from a higher SF and greater N in sand-dominant mixtures. However, the unconfined compressive strength of silt-dominant mixtures (SF > 30%) decreases with the increase in SF. In fine-dominant mixtures, the coarse particles can be dispersed by fine particles, which decreases the strength of coarse-fine mixtures [9,38,39]. Moreover, the unconfined compressive strength tends to decrease with the increase in N, despite the increase in ∆D_r_, owing to other more influential factors, such as dispersed sand particles and unfrozen water [10,40]. Therefore, ∆D_r_ significantly affects the strength of sand-dominant mixtures, but the addition of fine particles such as silt and repetitive loading could reinforce the strength of these mixtures.

### 4.3. Unconfined Compressive Strength vs. Volumetric Unfrozen Water Content

To clarify the effect of unfrozen water on the strength of silt-dominant mixtures (SF > 30%), the unconfined compressive strength versus θ_u_ is plotted in Figure 14. Figure 14 shows that the unconfined compressive strength of silt-dominant mixtures tends to decrease with the increase in N as the SF increases, while that of sand-dominant mixtures (SF ≤ 30%) increases as the N and SF increase. For silt-dominant mixtures, the unconfined compressive strength decreases with the increase in SF because the unfrozen water increases as the SF increases due to the effect of the physicochemical characteristics of silt particles [34,35]. It should be noted that unfrozen water may reduce the strength of frozen soils owing to the role of the lubricant [41]. In addition, the unconfined compressive strength tends to decrease with the increase in N because more unfrozen water may remain on the soil surface as repetitive loading reduces the number of voids [42,43]. However, the unconfined compressive strength of the sand-dominant mixture increases as the SF and N increase, which induce an increase in the unfrozen water content. Despite the increase in θ_u_, the effect of ∆D_r_ prevailed on the unconfined compressive strength (see Figure 13) for sand-dominant mixtures. Thus, the unfrozen water considerably reduced the strength of silt-dominant mixtures, and the addition of fine particles and repetitive loading should be avoided for silt-dominant mixtures.

The ∆D_r_ influences the strength of sand-dominant mixtures (SF ≤ 30%), whereas the θ_u_ and dispersing effect influences the strength of silt-dominant mixtures (SF > 30%). Thus, the addition of silt and repetitive loading is beneficial for sand-dominant mixtures but disadvantageous for silt-dominant mixtures. For the characterization of sand–silt mixtures that are likely to undergo cyclic freezing-thawing-repetitive loading, the effects of N at different SFs should be considered. Further, to broaden the estimation and prediction of the behavior of granular materials, numerical studies that consider particle interaction [44] and capillary force induced by water [45] can be utilized to verify the effects of ∆D_r_ and θ_u_ estimated in this experimental study.

## 5. Conclusions

The objective of this study is to estimate the effects of cyclic freezing-thawing-repetitive loading on the unconfined compressive strength of sand–silt mixtures with various silt fractions (SFs). The sand–silt mixtures continuously and repeatedly undergo cyclic freezing-thawing-repetitive loading, and the number of repetitive loading (N) is subjected up to 1000 in order to simulate a number of vehicle loads. During the cyclic freezing-thawing-repetitive loading, linear variable displacement (LVDT) and time domain reflectometry (TDR) sensors monitor the volumetric change and volumetric unfrozen water content (θ_u_). This enables the understanding of the influences on the strength of specimens. Sand–silt mixtures were prepared with SFs from 0 to 100% by weight (W_silt_/W_sand_ × 100%). The relative density (D_r_) and degree of saturation (S) were fixed at 60% and 15%, respectively. The sand–silt mixtures were placed in the floating oedometer cell in the repetitive loading system in the freezing chamber. The sand–silt mixtures underwent three cyclic freezing-thawing-repetitive loadings with N of 1, 100, and 1000. The specimen height and volumetric water content were measured during the cyclic freezing-thawing-repetitive loading using the LVDT and TDR probe. After three cyclic freezing-thawing-repetitive loadings, the sand–silt mixture was frozen and a uniaxial compression test was carried out. The measured unconfined compressive strength was compared with the change in relative density (∆D_r_) and θ_u_ to determine their influence on sand-dominant and silt-dominant mixtures. The main observations are as follows:

The change in the height of sand–silt mixtures after three cyclic freezing-thawing-repetitive loadings decreases as the SF increases up to 30% and increases beyond SF of 30%. However, ∆D_r_ should be considered because the range of the minimum to maximum void ratios varies with respect to the SF. The ∆D_r_ of sand–silt mixtures, which significantly affects the soil strength rather than void ratio in mixtures, increases with the increase in SF up to 30% and decreases beyond SF of 30%; this trend is opposite to that of the change in the height. As the volumetric water content varies during cyclic freezing-thawing-repetitive loading owing to the volume contraction, θ_u_ increases with the increase in SF and N because the pore water in small voids is easily affected by the physicochemical characteristics of the soil.
For the sand-dominant mixtures (SF ≤ 30%), the θu and ∆Dr increase with the increase in SF owing to the effect of the physicochemical characteristics of particles and a narrower range of the minimum and maximum void ratios, respectively. Furthermore, θu and ∆Dr increase with the increase in N because the unfrozen water remained in the small voids. By contrast, for the silt-dominant mixtures (SF > 30%), θu increases as SF increases because of the fine particles, while ∆Dr decreases or is constant because the small voids induce volume expansion during freezing, which prevents further ∆Dr. The N causes small voids before freezing, which generates more volume expansion during freezing that may decrease ∆Dr. Thus, the θu of silt-dominant mixtures subjected to N increases as ∆Dr decreases, while undergoing only a single freezing–thawing, i.e., N = 1, increases as ∆Dr is constant.The unconfined compressive strength of sand-dominant mixtures (SF ≤ 30%) increases with the increase in ∆Dr owing to the active particle contacts in the denser sand skeleton. Thus, the addition of SF and repetitive loading that stimulates ∆Dr reinforces the strength of sand-dominant mixtures. However, the unconfined compressive strength of silt-dominant mixtures (SF > 30%) decreases with the increase in θ_u_, which may reduce the strength of frozen soils owing to the role of the lubricant. The unconfined compressive strength of silt-dominant mixtures decreases with the increase in N owing to the dispersion of sand particles. Therefore, the addition of SF and repetitive loading, which increases the amount of unfrozen water and disperses the sand particles in the silt-dominant mixture, may reduce the strength of the soil.

## Figures and Tables

**Figure 1 sensors-20-05381-f001:**
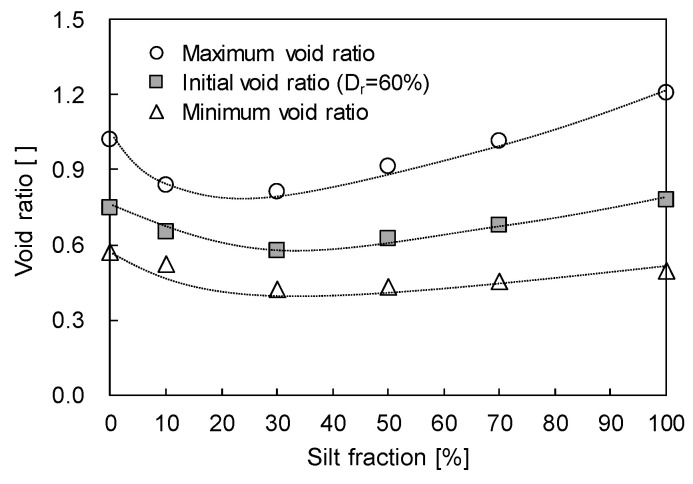
Maximum, minimum, and initial void ratios of sand–silt mixtures with respect to silt fraction. D_r_ denotes the relative density of specimen.

**Figure 2 sensors-20-05381-f002:**
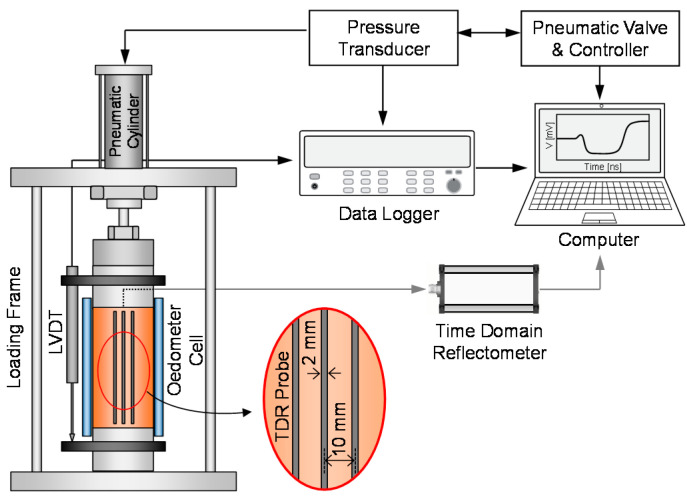
Schematic drawing of repetitive loading system. Linear variable displacement transducer (LVDT) and time-domain reflectometry (TDR) denote linear variable displacement transducer and time domain reflectometry, respectively.

**Figure 3 sensors-20-05381-f003:**
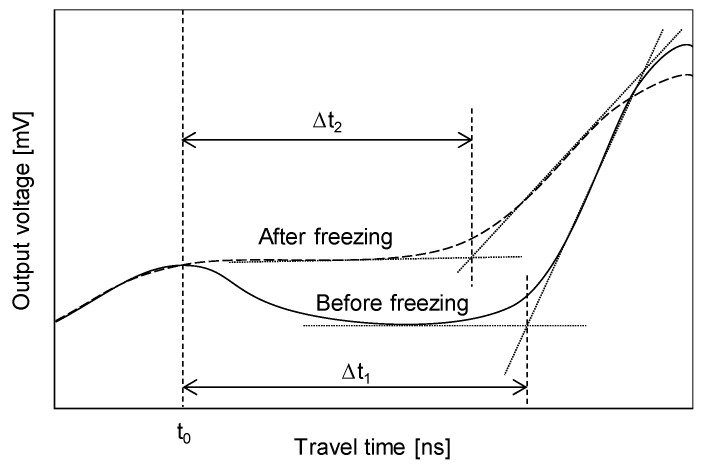
Conceptual TDR signals for soils before and after freezing.

**Figure 4 sensors-20-05381-f004:**
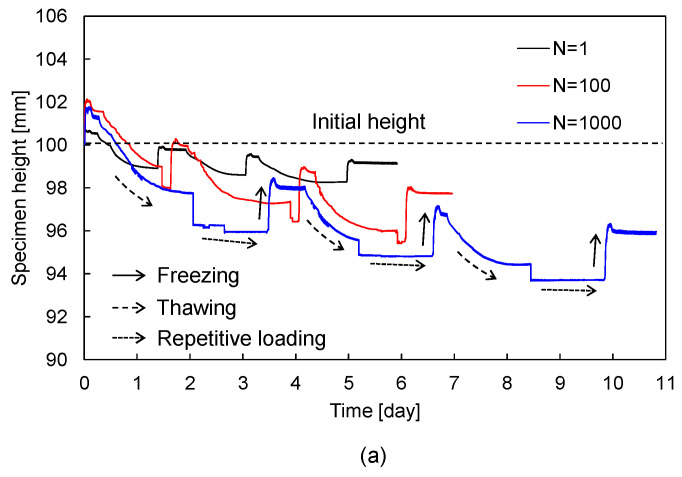
Variation of specimen height for SF = 70% mixture during: (**a**) cyclic freezing-thawing-repetitive loading under N = 1, 100, and 1000; (**b**) repetitive loading of N = 100 at 1st cycle. N denotes the number of repetitive loading.

**Figure 5 sensors-20-05381-f005:**
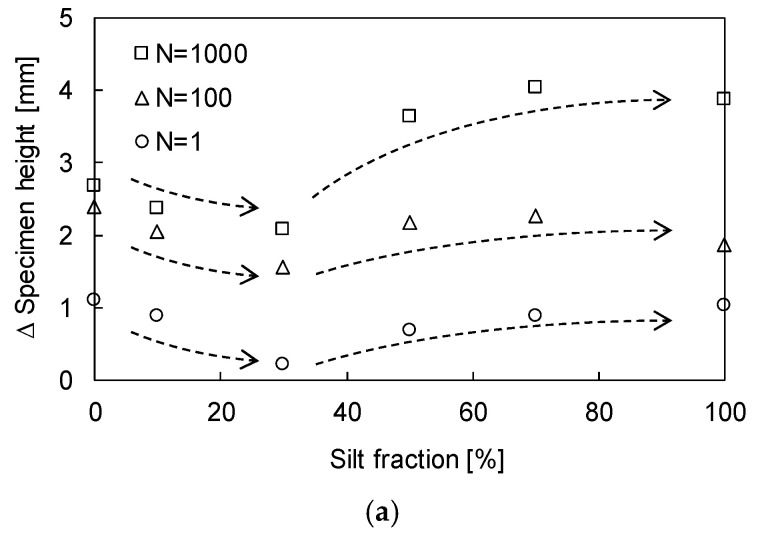
Volumetric change in sand–silt mixtures subjected to N = 1, 100, and 1000: (**a**) ∆Specimen height according to SF; (**b**) ∆Specimen height according to initial void ratio; (**c**) ∆D_r_ according to SF.

**Figure 6 sensors-20-05381-f006:**
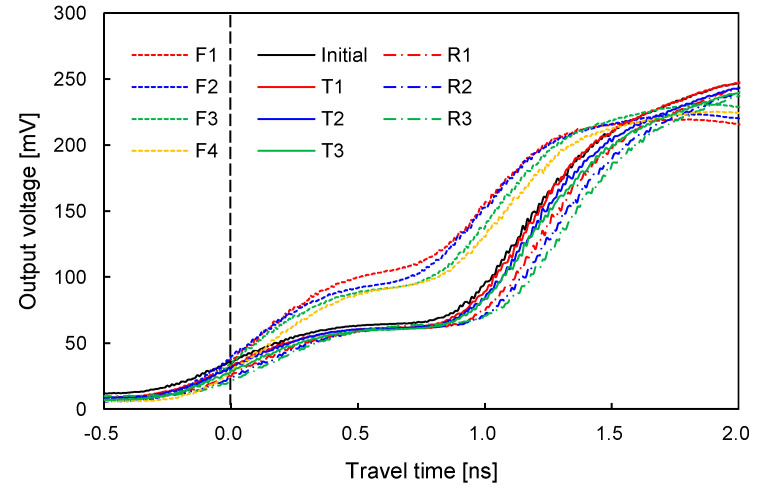
Typical signals of TDR during freeze-thaw cycles of sand–silt mixture at SF = 70% and N = 1000. Fn, Tn, and Rn denote the freezing (F), thawing (T), and repetitive loading (R) at nth cycle, respectively.

**Figure 7 sensors-20-05381-f007:**
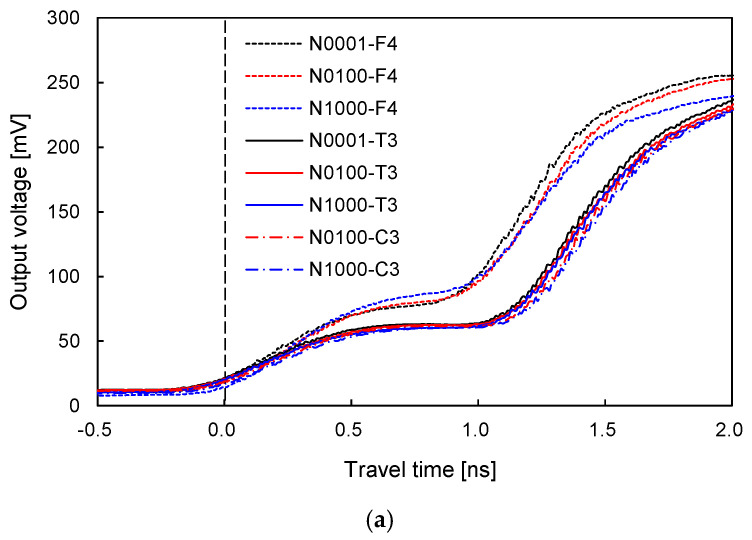
Typical signals of TDR of sand–silt mixtures subjected to N = 1, 100, and 1000: (**a**) SF = 0%; (**b**) SF = 100%. Nn denotes the number of repetitive loading.

**Figure 8 sensors-20-05381-f008:**
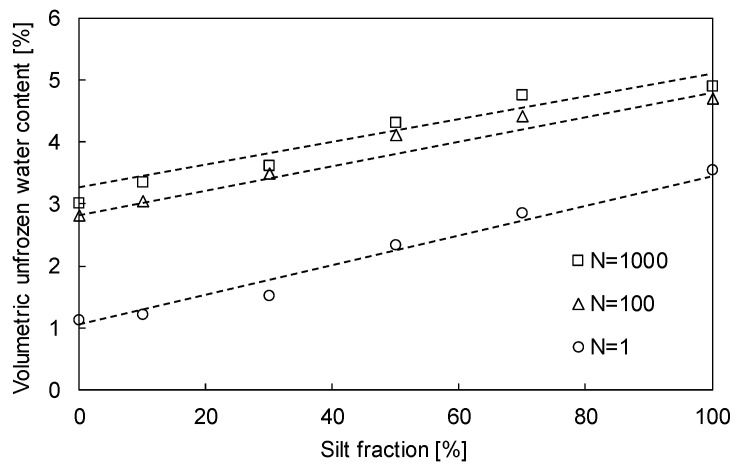
Volumetric unfrozen water content of sand–silt mixtures subjected to N =1, 100, and 1000 according to SF.

**Figure 9 sensors-20-05381-f009:**
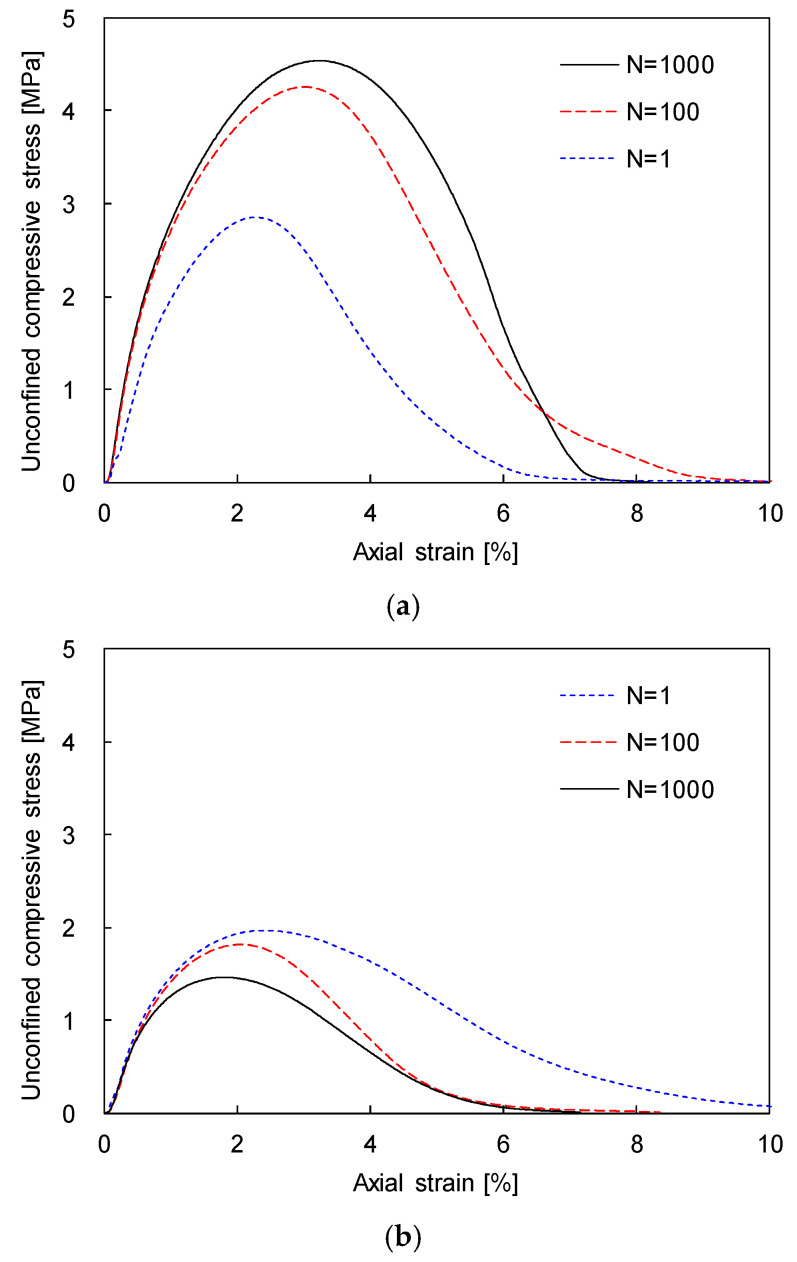
Typical unconfined compressive stress-strain curve of sand–silt mixtures subjected to N = 1, 100, and 1000: (**a**) SF = 30%; (**b**) SF = 70%.

**Figure 10 sensors-20-05381-f010:**
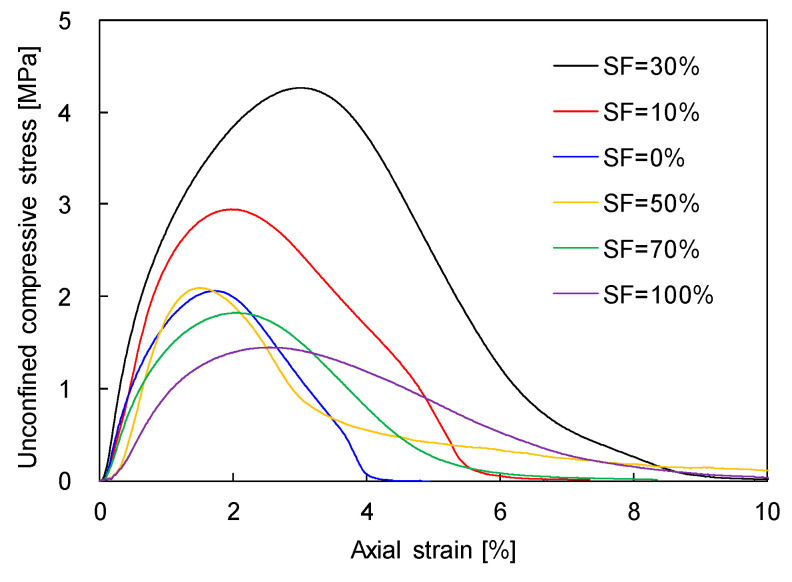
Typical unconfined compressive stress-strain curve with different SFs subjected to N = 100.

**Figure 11 sensors-20-05381-f011:**
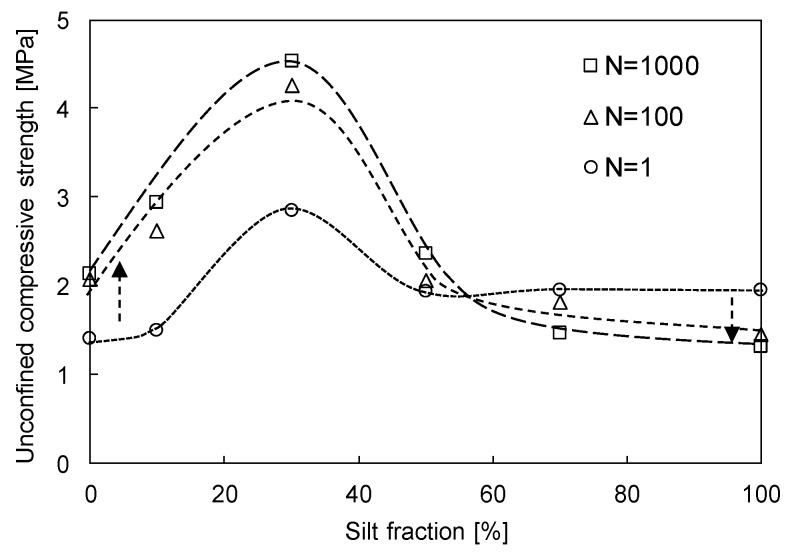
Variation of unconfined compressive strength versus SF of sand–silt mixtures subjected to N = 1, 100, and 1000.

**Figure 12 sensors-20-05381-f012:**
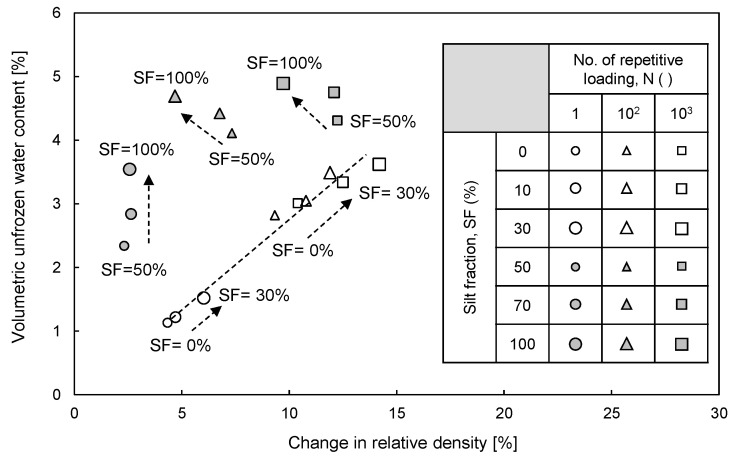
Relationship between volumetric unfrozen water content and change in relative density of sand–silt mixtures.

**Figure 13 sensors-20-05381-f013:**
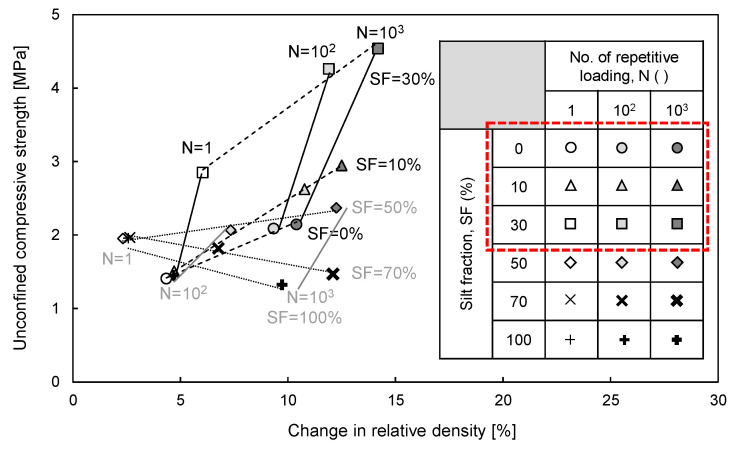
Relationship between unconfined compressive strength and change in relative density of sand–silt mixtures.

**Figure 14 sensors-20-05381-f014:**
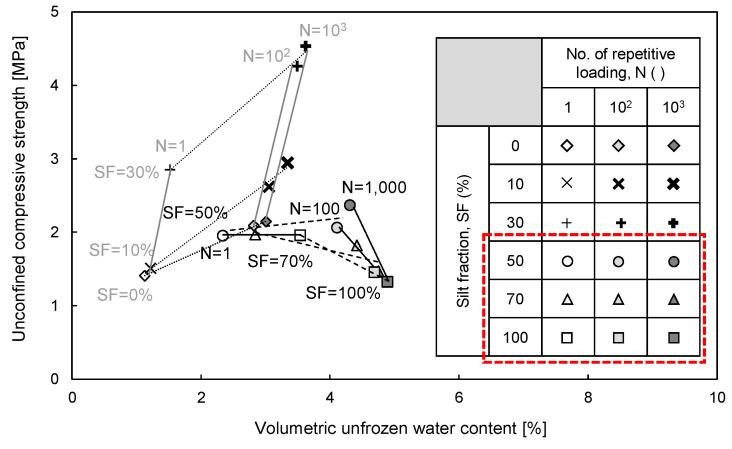
Relationship between unconfined compressive strength and volumetric unfrozen water content of sand–silt mixtures.

**Table 1 sensors-20-05381-t001:** Test procedure with abbreviations.

	Specimen Preparation	Freezing (Fn)	Thawing (Tn)	Repetitive Loading (Rn)	Uniaxial Compression Test
1st cycle	① Initial	② F1	③ T1	④ R1	-
2nd cycle	-	⑤ F2	⑥ T2	⑦ R2	
3rd cycle		⑧ F3	⑨ T3	⑩ R3	
4th cycle		⑪ F4	-		⑫ UCS

• Fn, Tn, and Rn denote freezing, thawing, and repetitive loading at nth cycle, respectively. UCS denotes the unconfined compressive strength. The number in circle presents the test procedure in order. The numbers of repetitive loading are 1, 100, and 1000 for each specimen.

**Table 2 sensors-20-05381-t002:** Coefficients of cubic polynomial relationships between volumetric unfrozen water content and relative permittivity.

Relationship	*a*	*b*	*c*	*d*
Kim et al. [15]	0.79	−13.95	84.30	−153.70
Kim et al. [26]	0.25	−3.70	25.49	−43.13
Lee et al. [27]	0.11	−2.31	19.66	−33.41

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
