# Peer review of "Strength Characteristics of Sand–Silt Mixtures Subjected to Cyclic Freezing-Thawing-Repetitive Loading"

_sensors, 2020, doi:10.3390/s20185381_

Round 1
Reviewer 1 Report
This paper assesses the unconfined compressive strength of sand-silt mixtures with silt fractions ranging from 0% to 100% subjected to cyclic freezing-thawing-repetitive loading conditions. In particular, the authors compare the variations of the unconfined compressive strength with a change in relative density and volumetric unfrozen water content estimated by time domain reflectometry (TDR) sensor to discuss the effect of the number of repetitive loading and unfrozen water. In general, this paper is well organized, and the experimental concept is applicable for the readers of Sensors journal. However, reviewer suggests several comments to improve the clarity of the manuscript. Specific comments follow: 1) 2. Materials and Methods. Reviewer recommends to provide a freezing temperature and the reason of the selected target temperature. 2) Line 71-73 and Figure 1. How was the maximum void ratio for sand-silt mixture with silt fraction beyond 15% determined without segregation? 3) Line 107. How did the authors determine a frequency of repetitive loading at 0.083 Hz without considering the probable influence of loading frequency? Specifically, the authors should describe the influence of loading frequency on the volume change for each specimen with different number of repetitive loading. 4) Line 154-156 and Figure 4(b). Describe a physical meaning of the strain convergence at the number of repetitive loading of 100. Are there similar observations in previous studies? 5) Line 248. Did the authors consider probable change in fines content due to the particle crushing during repetitive loading? Provide the evidence if the particle crushing effect is ignored.Author Response
The authors are grateful to the reviewers for their valuable comments.
Please see the attachment.

Reviewer 2 Report
This paper is providing interesting data on the strength characteristics and micromechanics of sand-silt mixtures, when these are subjected to cyclic freezing-thawing-repetitive loading. The English is acceptable, but the writing style should be improved by the authors in their future papers, because it causes confusion, and it also demands from the reader great effort to read and re-read the paper many times, from the beginning, to finally understand what the authors want to say. There is too much text to explain some details, while the most relevant information is, in some parts, not revealed to the reader. The symbols (SF, theta_u, etc.) are defined in the beginning but their meaning is not recalled after several pages, which makes the reading difficult. Therefore, the authors should improve this aspect, following my specific recommendations (see below), and take additional comments (see below) in the revision of this manuscript into account, before I can recommend acceptance.
1) Parameter definitions: It would be helpful if the authors could add, somewhere in the paper, a table with 3 columns, so that the reader can check it whenever a quantity is mentioned in the paper: parameter symbol unit silt-fraction SF % (...)
2) Figure 1: It is impossible for the reader to understand what this figure is actually showing, unless the reader is a co-author or knows a very similar paper where a similar figure has been shown. How many samples do you have for each value of the silt fraction? Only 1 semple per silt fraction? If yes, then what do you mean with "maximum" and "minimum" void ratio? Do you measure the void ratio at different positions in the sample, or do you measure one single void ratio for the sample, but have various samples to take an average (in this case, where are the error bars?)? Could you please describe in more detail when, approximately, the maximum and the minimum void ratio values occur during the experiment? In summary: The authors should add a new paragraph where these questions are answered explicitly.
3) Line 130: "Previous studies have revealed that the relationship between the volumetric water content and relative permittivity can be expressed by a cubic equation [15, 21, 22]." Is this cubic equation too big to be mentioned in the paper? It would be helpful to display it, so that the reader can make a comparison between it and your results. Moreover, you should make this comparison explicitly, i.e., in some part of the paper you should explicitly write whether your results agree with or contradict the cubic equation.
4) Line 137: What is theta_u? Please include the definition of theta_u again in the sentence.
5) Figure 3: Please tell that this has been obtained from your measurements, or has this figure been taken from another work? In this case, please include the citation.
6) Line 147: What is "SF"? Please include the definition of SF again in the sentence.
7) Line 172: "because the mechanical characteristics of sand–silt mixtures, such as strength, are significantly affected by the Dr rather than the void ratio". This sentence is confusing because if Dr is the density (in %), then the void ratio (which I understand as "porosity") is equal to 1 - Dr. Alternatively, if Dr is the density in g/cm3, then the void ratio is equal to 1 - Dr/Dmaterial, where Dmaterial is the density of a single particle. Therefore, both quantities void ratio and Dr are directly related. How can the mechanical characteristics be then significantly affected by one rather than the other quantity, if both quantities are perfectly complementary to each other? Or is the void ratio something different from the porosity? Please also include again the definition of Dr in the sentence.
8) Figure 6: It is better to use dotted lines for the F1, F2, F3, and F4 curves, to better differentiate them from the R... curves.
9) Figure 7a, 7b: It is better to use dotted lines for the F1, F2, F3, and F4 curves, to better differentiate them from the C... curves.
10) Section conclusions: The most attentive reader will have difficulties to remember the meaning of all symbols that appear in the text of this section. Please recall their meaning, alternatively refer to a table where all symbols are explained (see my comment 1 above).
11) Section conclusions: Compared to previous studies, please cite 2-3 main novelties presented in your manuscript.
12) The weak part of this paper is that it doesn't connect well the experimental framework to the community of numerical simulation experts. Your results are really very important to improve particle-based models. In particular, great progress is being made in the model and application of granular materials with relevance for the industry, but your experiments show that the micromechanics of granular materials can change considerably depending on the boundary conditions to which they are subjected (and the level of polydispersity). Therefore, it would be pertinent to acknowledge the excellent progress in the particle-based simulation, e.g., by citing these papers: Wet granular flow control through liquid induced cohesion A Jarray, V Magnanimo, S Luding Powder technology 341, 126-139 (2019) Attractive particle interaction forces and packing density of fine glass powders EJR Parteli, J Schmidt, C Blümel, KE Wirth, W Peukert, T Poeschel Scientific Reports 4, 6227 (2014) and stress that it would be interesting to incorporate the insights presented in your paper into the particle-based models such as the ones of the papers above, to make these models stronger and better applicable to a broader range of environmental and industrial applications. Such a remark, including the suggested citations of particle-based simulations, would help to enhance the visibility of the article and also to connect the communities of experimentalists and modelers in this field that still demands so much research and (most importantly) the combined efforts of both communities.
Author Response
The authors are grateful to the reviewers for their valuable comments.
Please see the attachment.

Round 2
Reviewer 2 Report
The authors revised the paper satisfactorily.
I now recommend acceptance for publication in Sensors.